# Genetic Ablation of MiR-22 Fosters Diet-Induced Obesity and NAFLD Development

**DOI:** 10.3390/jpm10040170

**Published:** 2020-10-14

**Authors:** Monika Gjorgjieva, Cyril Sobolewski, Anne-Sophie Ay, Daniel Abegg, Marta Correia de Sousa, Dorothea Portius, Flavien Berthou, Margot Fournier, Christine Maeder, Pia Rantakari, Fu-Ping Zhang, Matti Poutanen, Didier Picard, Xavier Montet, Serge Nef, Alexander Adibekian, Michelangelo Foti

**Affiliations:** 1Department of Cell Physiology and Metabolism, Faculty of Medicine, University of Geneva, 1206 Geneva, Switzerland; Monika.Gjorgjieva@unige.ch (M.G.); Cyril.Sobolewski@unige.ch (C.S.); a.sophie.ay@gmail.com (A.-S.A.); Marta.Sousa@unige.ch (M.C.d.S.); Dorothea.portius@gmail.com (D.P.); flavienberthou@hotmail.com (F.B.); margot.fournier@unige.ch (M.F.); christine.maeder@unige.ch (C.M.); 2Department of Chemistry, The Scripps Research Institute, 130 Scripps Way, Jupiter, FL 33458, USA; DAbegg@scripps.edu (D.A.); aadibeki@scripps.edu (A.A.); 3Institute of Biomedicine, Research Centre for Integrative Physiology and Pharmacology, and Turku Center for Disease Modeling, University of Turku, FI-20014 Turku, Finland; piaranta@utu.fi (P.R.); fuping.zhang@helsinki.fi (F.-P.Z.); matti.poutanen@gu.se (M.P.); 4Department of Cell Biology, Faculty of Science, University of Geneva, 1205 Geneva, Switzerland; didier.picard@unige.ch; 5Department of Radiology, Faculty of Medicine, University of Geneva, 1206 Geneva, Switzerland; xmontet@infomaniak.ch; 6Department of Genetic Medicine and Development, Faculty of Medicine, University of Geneva, 1206 Geneva, Switzerland; Serge.Nef@unige.ch; 7Diabetes Center, Faculty of Medicine, University of Geneva, 1206 Geneva, Switzerland

**Keywords:** fatty liver disease, microRNAs, lipid metabolism, glycolysis, obesity

## Abstract

miR-22 is one of the most abundant miRNAs in the liver and alterations of its hepatic expression have been associated with the development of hepatic steatosis and insulin resistance, as well as cancer. However, the pathophysiological roles of miR-22-3p in the deregulated hepatic metabolism with obesity and cancer remains poorly characterized. Herein, we observed that alterations of hepatic miR-22-3p expression with non-alcoholic fatty liver disease (NAFLD) in the context of obesity are not consistent in various human cohorts and animal models in contrast to the well-characterized miR-22-3p downregulation observed in hepatic cancers. To unravel the role of miR-22 in obesity-associated NAFLD, we generated constitutive *Mir22* knockout (miR-22KO) mice, which were subsequently rendered obese by feeding with fat-enriched diet. Functional NAFLD- and obesity-associated metabolic parameters were then analyzed. Insights about the role of miR-22 in NAFLD associated with obesity were further obtained through an unbiased proteomic analysis of miR-22KO livers from obese mice. Metabolic processes governed by miR-22 were finally investigated in hepatic transformed cancer cells. Deletion of *Mir22* was asymptomatic when mice were bred under standard conditions, except for an onset of glucose intolerance. However, when challenged with a high fat-containing diet, *Mir22* deficiency dramatically exacerbated fat mass gain, hepatomegaly, and liver steatosis in mice. Analyses of explanted white adipose tissue revealed increased lipid synthesis, whereas mass spectrometry analysis of the liver proteome indicated that *Mir22* deletion promotes hepatic upregulation of key enzymes in glycolysis and lipid uptake. Surprisingly, expression of miR-22-3p in Huh7 hepatic cancer cells triggers, in contrast to our in vivo observations, a clear induction of a Warburg effect with an increased glycolysis and an inhibited mitochondrial respiration. Together, our study indicates that miR-22-3p is a master regulator of the lipid and glucose metabolism with differential effects in specific organs and in transformed hepatic cancer cells, as compared to non-tumoral tissue.

## 1. Introduction

Obesity and the metabolic syndrome are associated with a spectrum of hepatic pathologies classically defined as non-alcoholic fatty liver disease (NAFLD) or, as suggested more recently, as metabolic dysfunction-associated fatty liver disease (MAFLD). These hepatic metabolic disorders start with an excessive accumulation of lipids in the hepatocytes (steatosis). Chronic accumulation of fat in hepatocytes disrupts with time the normal hepatic homeostasis through abnormal molecular signaling, mitochondrial dysfunction, endoplasmic reticulum (ER) stress, and hepatocyte death. These insults elicit, on a long term, a chronic inflammatory response leading to non-alcoholic steatohepatitis (NASH) and favoring hepatic fibrogenesis. This pathological state can then further progress to cirrhosis, a disease characterized by the presence of regenerating hepatocytes nodules, an extended fibrosis and portal hypertension. An end-stage of these hepatic disorders is the further development of hepatocellular carcinoma (HCC), which mostly arises in cirrhotic livers, but which can also occur with NASH/fibrosis only [1,2].

A wealth of data indicates that epigenetic mechanisms contribute importantly to the progression of NAFLD towards severe stages and HCC [3]. In particular, several microRNAs (miRNAs), whose expression and/or activity are altered in NAFLD, have been identified as key factors involved in these metabolic disorders and liver cancer [3,4]. Through their ability to bind to the 3′-UTR (3′-untranslated region) of messenger RNAs (mRNA), miRNAs mediate mRNA decay or translation inhibition, thereby representing key post-transcriptional regulators of many genes involved in metabolism, cellular stress, and carcinogenesis. As miRNAs can be used as therapeutic targets, intense efforts are being devoted to identify and characterize miRNAs deregulated with NAFLD/NASH and cancer in order to find novel diagnostic and/or therapeutic tools.

Of particular interest for hepatic diseases is miR-22-3p, since it is one of the most abundantly expressed miRNAs in the liver [5,6]. However, the pathophysiological role of miR-22-3p in NAFLD/NASH and cancer is still unclear, with contrasting findings reported to date. Previous studies have highlighted an upregulation of miR-22-3p in steatotic livers of db/db mice, wild type (WT) mice fed a highly caloric Western diet or with alcohol-induced steatosis [7,8] and upregulated circulating levels of miR-22-3p in patients with NASH [9]. The analysis of Western diet-induced obese mice suggested a direct regulation of miR-22 on fibroblast growth factor receptor 1 (*FGFR1*) expression. Intriguingly, this regulation no longer took place when mice were treated with obeticholic acid, as both miR-22 and *FGFR1* were found upregulated during treatment with this bile acid resulting in steatosis alleviation [8]. In db/db mice and rats, this miRNA was shown to promote hepatic gluconeogenesis [7,8,9,10] and loss of miR-22 in mice was suggested to prevent hepatic steatosis and inflammation development following a high-calorie diet intake [11]. In contrast, miR-22-3p expression was shown to be reduced in hepatic tissues of rats spontaneously developing diabetes [12] and to increase the triglyceride content in HepG2 hepatoma cells [13]. Thus, although miR-22-3p seems to play an important role in hepatic metabolism, its precise role remains yet to be fully deciphered.

Strikingly, miR-22-3p expression was repeatedly reported to be downregulated in mouse and human HCC [14,15] and shown to target oncogenic factors such as histone deacetylase 4 (*HDAC4*), ezrin (*EZR*), and basigin (*CD147*) [14,15,16]. These observations led to consider this miRNA as a tumor suppressor and a potential therapeutic target for HCC. Metabolic switches in cancer cells, i.e., Warburg effects, are key drivers in hepatic carcinogenesis [17], and whether miR-22-3p affects cancer cell metabolism in the same manner as in non-tumoral tissues remains to be established. This is of key importance before considering miR-22-3p as a therapeutic target since increasing evidence indicates that miRNA activities might be highly dependent on cellular stress, environmental factors and changes in cellular signaling and gene expression/activity associated with cell transformation [18].

In this study, we addressed these questions by investigating metabolic alterations in obesity-associated NAFLD in vivo using a newly generated *MiR22* knockout mouse model, and in well-characterized liver cancer cells.

## 2. Materials and Methods

### 2.1. Reagents and Antibodies

All reagents and diet compositions are described in the Appendix A section. RNAs isolated from human liver biopsy specimens of obese patients displaying or not steatosis were previously described [19].

### 2.2. Maintenance of Constitutive Mir-22 Knockout (miR-22KO) Mice

Male *Mir22* knockout (miR-22KO) and control (CTL) littermate mice were fed with a high fat diet (HFD: 60 kJ% fat), or a standard chow diet (CD: 11 kJ% fat), for 12 weeks (diet composition and origin is described in Appendix A). At the end of the study, mice were anesthetized by isoflurane and sacrificed by decapitation and blood and organs were collected and weighted. The generation of transgenic mice is described in the Appendix A section. Animal phenotyping and functional characterizations, as well as the histological/molecular analyses of explanted tissues are detailed in Appendix A. Animal care and experimental procedures were performed in accordance with the Swiss guidelines for animal experimentation and ethically approved by the Geneva Health Head Office (ethic approval code: CER. 06-156 (NAC 06-049).

### 2.3. Proteomic Analysis

LC-MS/MS analysis was performed on explanted liver tissues from CTL and miR-22KO mice fed a HFD diet for 12 weeks. MS data were then analyzed with MaxQuant (V1.5.2.8) and searched against the mouse proteome. A detailed description of the proteomic analysis is provided in the Appendix A section.

### 2.4. Seahorse Analysis

A total of 200.000 Huh7 cells per well were seeded in 6-well plates and 24 h later were transfected with 20 nM of control or miR-22-3p oligonucleotides (Dharmacon) using Viromer blue transfection reagent. At 24 h after transfection, 25.000 cells/well were re-plated in a 96-well Seahorse plate and incubated overnight at 37 °C. At 48 h post-transfection, cells were analyzed either with MitoStress kit or GlycoStress kit, as recommended by the Seahorse Agilent guidelines.

### 2.5. Statistical Analysis

Data are expressed as mean ± standard error of the mean (SEM). Statistical significance between groups was assessed by unpaired Student’s *t* test with *p* < 0.05 or analysis of variance (ANOVA, one or two way followed by a Dunnett’s multiple comparisons test). For the proteomic analysis, *p*-values were calculated by a Student’s t-test. Adjusted p values were calculated by the Benjamini and Hochberg method. *p*-values < 0.05 (*) were considered statistically significant.

## 3. Results

### 3.1. Mir-22-3p Expression in the Liver is Not a Reliable Marker of NAFLD and/or NASH.

MiR-22-3p is the fourth most abundant miRNA in the liver and is strikingly more expressed than its passenger strand, miR-22-5p (Appendix A). In order to assess the potential roles of miR-22-3p in metabolic disorders associated with NAFLD/obesity, we first cross-referenced common predicted/validated miR-22-3p targets between mice and human, which were further compared with gene lists associated to glucose and lipid metabolism (Figure 1A,B). This in silico analysis revealed 123 potential gene targets of miR-22-3p associated with the glucose and/or lipid metabolism and Gene Ontology (GO) analyses further highlighted significant enrichments in various lipid catabolism and glucose metabolism processes (Figure 1C,D and Appendix A). Consistent with an important role of miR-22-3p in liver metabolism, GO analysis for tissue expression indicated a significant enrichment of potential miR-22-3p targets in the liver (Appendix A).

Downregulation of miR-22-3p expression was previously reported in various human cancers, including HCC [14,15,16,20,21]. However, whether miR-22-3p expression or activity is already impaired at early stages of liver disease characterized by the presence of steatosis is still unclear. We, therefore, analyzed publicly available transcriptomic Gene Expression Omnibus (GEO) datasets reporting miR-22-3p expression in human NAFLD (Figure 1E and Appendix A). Our data indicate a significant downregulation of miR-22-3p in one dataset (GSE49012), while no significant alteration was observed in the other (GSE59492). A significant decrease of miR-22-3p was further highlighted in patients with cholestasis and alcoholic hepatitis/cirrhosis in GSE49012 and GSE59492, respectively. We further investigated miR-22-3p expression in a small cohort of human liver biopsies from obese human patients having or not steatosis that we previously characterized [19]. In this cohort (n = 7–9 patients per group), hepatic miR-22-3p expression was significantly downregulated in the liver of human obese patients with steatosis, as compared to obese but non-steatotic individuals (Figure 1F). Similar discrepancies were found in several mouse models of hepatic steatosis available in our laboratory (see histological characterization of ob/ob, db/db and liver-specific phosphatase and tensin homologue (*Pten*) knockout (LPTENKO) mice in Appendix A), as miR-22-3p is significantly induced in the liver of ob/ob mice, while unchanged in db/db mice and downregulated in mice having steatosis following deletion of *Pten* specifically in hepatocytes (LPTENKO mice) (Figure 1G).

With liver fibrosis or diabetes, mouse transcriptomic datasets indicate a significant decrease of miR-22-3p in only one dataset (GSE77271), while other studies did not reveal any significant change (Figure 1H, and Appendix A). In support of miR-22-3p downregulation with hepatic steatosis and/or inflammation, exposure of mouse primary hepatocytes (MPH), SK-Hep1 cells or human primary hepatocytes (HPH) to free fatty acids (e.g., oleate, palmitate or linoleate) ± inflammatory mediators (TNFα) also significantly reduced miR-22-3p expression (Appendix A).

Together, these data suggest that although miR-22-3p may represent a critical regulator of hepatic metabolism, its expression, without information about its activity, does not represent a reliable biomarker for liver metabolic disorders.

### 3.2. Mir22 Deletion Worsens Diet-Induced Obesity

To investigate the role of miR-22-3p in metabolic homeostasis, mice bearing a constitutive deletion of *Mir22* were engineered (Appendix A, Appendix A) by crossing *Mir22* floxed mice with mice expressing germ-cell specific *Ngn3* promoter-driven transgenic Cre. The constitutive deletion of *Mir22* (miR-22KO mice) was assessed by RT-qPCR in different tissues. The genomic deletion resulted in undetectable levels of mature miR-22-3p in tissues including the liver, epididymal white adipose tissue (eWAT) and gastrocnemius (Appendix A). In order to assess the functional relevance of miR-22 loss in obesogenic conditions, miR-22KO mice were submitted to either chow diet (CD, 12 weeks), or high-fat containing diet (HFD, 12 weeks, see Appendix A) [22]. Diet specifications are described in Appendix A section, Appendix A. Under CD, miR-22KO mice did not display any obvious phenotype regarding their weight, adiposity, steatosis and plasma lipid levels, except an increase in plasma triglycerides (Figure 2A–D). However, when mice were challenged with a HFD for 12 weeks, *Mir22* deletion induced a faster and higher weight gain as compared to CTL mice (Figure 2A). Alanine transaminase (ALAT) levels were also increased in miR-22KO mice challenged with HFD, indicating that hepatocellular damages occur with *Mir22* deficiency, but no differences were observed in circulating levels of cholesterol derivatives, triglycerides and NEFAs, compared to WT HFD littermates (Figure 2B,C). The increased weight gain was associated with a significant expansion of the fat mass as assessed by *(i*) EchoMRI analysis *(ii),* the weight of epididymal and mesenteric fat in explanted adipose tissues and (*iii*) CT-Scan analysis of the volume of subcutaneous and intra-abdominal fat (Figure 2D and Appendix A). Finally, fatness of miR-22KO mice under HFD was not linked to a significant decrease in energy expenditure, as indicated by indirect calorimetry evaluating the respiratory exchange rate, food consumption, water consumption, fuel oxidation and locomotor activity (Appendix A). Lipid trafficking in epidydimal white adipose tissue (eWAT) remained unchanged compared to CTL mice, as we observed no difference in cluster determinant 36 (CD36) levels (Appendix A). However, by analyzing key lipogenic enzymes in the eWAT, we observed an increase in protein levels of the enzymes FAS (fatty acid synthase), ACC (acetyl CoA carboxylase) and SCD1 (stearoyl-CoA desaturase 1), suggesting an increase in lipid synthesis in the absence of miR-22 in this organ (Appendix A). Finally, we did not observe an impact on insulin sensitivity of the eWAT, as phosphorylation of the insulin receptor (IR) and its downstream effector AKT was unaltered in explanted eWAT following insulin stimulation prior to mouse sacrifice (Appendix A).

### 3.3. Mir22 Deletion in Mice Fosters Hepatic Steatosis and Glucose Intolerance

In miR-22KO mice fed CD, we did not observe any significant anomalies of liver morphology/weight as compared to CTL mice (Figure 3A and Appendix A). However, when mice were fed 12 weeks with a HFD, miR-22KO mice developed strong hepatomegaly and extended hepatic steatosis, whereas CTL mice developed only a modest increase in liver weight and steatosis under these conditions (Figure 3A,B and Appendix A).

Since diet-induced-obesity and hepatic steatosis are frequently associated with glucose intolerance, insulin resistance and hyperinsulinemia, we then performed glucose tolerance tests (GTT) in mice fed CD or HFD for 12 weeks. Glucose intolerance was exacerbated by miR-22-3p deficiency in mice fed both CD or HFD, indicating an obesity-independent control of miR-22-3p on glucose tolerance (Figure 3C,D). The increased pic of plasma glucose at 15 min in miR-22KO mice under chow diet (Figure 3C), with then a similar rate, as compared to CTL mice, of glucose disappearance, further suggests that miR-22-deficiency leads to alterations in hepatic glucose output. Fasted insulinemia was however increased, as well as C-peptide levels in the serum (although not statistically significant) in miR-22KO mice fed HFD for 12 weeks, indicative of an increased insulin resistance triggered by miR-22-3p deficiency under these conditions (Figure 3E).

### 3.4. Proteomic Analysis of MiR-22KO Mice Liver Tissue Uncovers Alterations of Glycolysis and Hepatic Lipid Trafficking

To gain further insights into the molecular mechanisms by which miR-22-3p promotes hepatic steatosis, we performed a proteomic analysis of hepatic tissue extracts from CTL and miR-22KO mice previously challenged with the obesogenic diet for 12 weeks. A total of 2708 proteins were identified in the liver from diet-induced obese CTL and miR-22KO mice. Among those, 148 proteins were upregulated and 183 proteins were downregulated in the liver of miR-22KO mice (Figure 4A and Appendix A). Upregulated proteins, which are expected to be potential targets of miR-22 in miR-22KO mice, were then compared with a list of predicted mouse miR-22-3p targets (miRwalk database, Figure 4B), thereby highlighting 54 potential miR-22-3p target candidates. Gene Ontology (GO) analyses of these 54 candidates revealed a significant enrichment of factors involved in lipid/glucose metabolic processes (Figure 4C). Based on literature, 55% (30 proteins) of the identified candidates were associated with lipid/glucose metabolism, while the remaining 24 protein candidates are involved in other processes. Importantly, most candidates associated with metabolism are involved in lipogenesis, lipid catabolism, trafficking, and glycolysis (Figure 4D,E) and form a tightly interconnected network with protein-protein interaction and co-expression patterns (Figure 4F). At the mRNA level, *Cd36*, as well as other lipid transporters, which were not identified in the proteomic analysis (i.e., fatty acid transport protein 1 and 2—*Fatp1*, *Fatp2*), were also significantly increased (Figure 5A). However, mRNA levels of fatty acid synthase, acetyl-CoA carboxylase and stearoyl-CoA desaturase 1 (*Fasn*, *Acaca,* and *Scd1,* respectively) remained unchanged (Figure 5A) while an increase in corresponding protein expression was measured by proteomics (Figure 4D). These data suggest both miR-22-dependent degradation and translational blockade of specific transcripts by miR-22, thus supporting the importance of our proteomic approach instead of transcriptomic analysis to identify potential miR-22 targets involved in the hepatic metabolism. In addition to significantly altered lipid trafficking in the liver, we further observed an increased expression of the predicted miR-22-3p targets acyl-CoA dehydrogenase medium chain (*Acadm*), peroxisome proliferator activated receptor alpha *(Ppara),* and carnitine palmitoyltransferase 1A (*Cpt1a*) by RT-qPCR analyses, which are important factors promoting hepatic lipid catabolism (Appendix A). However, it cannot be excluded that the upregulation of these factors is a consequence of increased lipid uptake by the liver and not because of miR-22 deficiency per se.

Our proteomic analysis also revealed an increased protein expression of key factors involved in glycolysis (Figure 4E), which can contribute to the exacerbated hepatic steatosis observed in the absence of miR-22, by feeding substrate for lipid biosynthetic pathways. We could confirm increased mRNA levels of potential miR-22-3p targets uncovered in the proteomic analyses, i.e., enolase 1 (*Eno1*), sedoheptulokinase (*Shpk*), and glucokinase (*Gck*), suggesting that miR-22-3p regulates these factors by degrading their respective mRNAs (Figure 5B). mRNA expression of other glycolytic genes that were not predicted to be targets of miR-22-3p by the miRwalk software, or were not detected with the proteomic analysis, such as pyruvate kinase M2 (*Pkm2*) and phosphofructokinase, liver type (*Pfkl*), were also found increased in liver tissues of obese miR-22KO mice (Figure 5B). Together these data indicate that *MiR22* deficiency in the liver is associated with the upregulation of key factors promoting lipid uptake and glycolysis in the liver, thereby fostering the development of hepatic steatosis in obesogenic conditions.

### 3.5. Mir-22-3p Induces a Classic Warburg Effect in Hepatic Cancer Cell Line

Upregulation of glycolysis occurs in NAFLD and is one of the hallmarks of cancer cells required to sustain its development and progression towards malignancy [23,24,25]. Increased glycolysis with miR-22 deficiency is therefore potentially one of the mechanisms explaining the tumor suppressive activity that has been suggested for miR-22 in various cancers, including HCC [14,15,16]. To verify this hypothesis, we investigated miR-22-dependent glycolytic processes in hepatic cancer cells. We analyzed three different human hepatic cancer cell lines, Huh7, HepaRG, and HepG2 cells, and found in all of them very low expression levels of miR-22-3p, as compared to normal human primary hepatocytes (HPH) (Appendix A). Analysis of the miRMine database further confirmed miR-22-3p as one of the lowest expressed miRNA in human HepG2 cells (Appendix A). We further examined glycolytic rates (GlycoStress) by Seahorse analysis in Huh7 hepatic cancer cells transfected or not with miR-22-3p mimicking oligonucleotides (miR-22 mimics) for 48 h. Intriguingly, both glycolysis rate and capacity in cells expressing miR-22 mimics tend to increase (Figure 5C) with no changes in the mRNA expression of the glycolytic genes *Eno1* and *Shpk* (Appendix A), in contrast to what we observed in non-tumoral but steatotic livers of miR-22KO mice fed HFD (Figure 5B). Seahorse analysis using a MitoStress test under the same conditions further indicated a decrease in the oxygen consumption rates in Huh7 cells expressing miR-22 mimics with statistical differences in the basal and decoupled maximum respiration, ATP production, proton leakage and the non-mitochondrial respiration (Figure 5D).

Together, these analyses indicate that miR-22 expression in hepatic cancer cells such as Huh7 cells triggers a classical Warburg-like metabolic switch from mitochondrial respiration to glycolysis.

## 4. Discussion

In this study, we found that miR-22-3p plays an important role in hepatic metabolism, as well as in whole-body metabolism. In particular, our data demonstrate that genetic deletion of *Mir22* triggers body fat accumulation and an exacerbated hepatic steatosis in mice fed an obesogenic diet. At the mechanistic level, we showed that *Mir22* deficiency in vivo leads to an increased expression of lipogenic enzymes in the adipose tissue, increased hepatic expression of glycolytic and lipid trafficking genes, entailing an increased lipid accumulation in the liver. However, our in vitro data converge to different conclusions about the role of miR-22 in hepatic human cancer cells, since miR-22 expression led to a classical Warburg-like effect [26], by increasing glycolysis and decreasing mitochondrial respiration rates. These findings highlight an important role of miR-22 in the hepatic regulation of the lipid/glucose metabolism, but which paradoxically appears to be opposite in pre-cancerous lesions of the liver associated with fatty liver diseases, as compared to already transformed hepatic cancer cells.

Understanding the role and function of a specific miRNA in metabolic disorders is a complex issue due to the pleiotropic effects of a single miRNA on multiple targets depending on the tissue and the type of environmental stress. In addition, differences in mouse models and/or strains, as well as in the methodology used to modulate miRNA expression in vivo, can lead to significantly different conclusions regarding the function of a specific miRNA in metabolic processes [22,27,28]. In this regard, our findings are divergent from those in previous studies suggesting that miR-22 deficiency/inhibition in mice protects from dyslipidemia induced by highly caloric diets [8,9,10,11]. Although we do not have a clear explanation for these discrepant results, different mouse strains having different sensitivity to the obesogenic diets, the exact content of the administered diet, the experimental timeframe, murine genetic background, as well as the breeding conditions of the mice are all susceptible to alter the outcomes of *MiR22* deficiency in metabolic disorders [29,30,31].

Furthermore, genetically-induced obesity models such as db/db are also susceptible to yield discrepant results due to these variables [32,33]. Indeed, increased expression of miR-22-3p in the liver of db/db mice was previously reported and silencing of miR-22-3p expression in these mice by antagomir injection was shown to improve glucose, pyruvate, and insulin tolerance [7,8,9,10,11,12,13]. In contrast to this study, our analyses of db/db mice did not reveal increased expression of miR-22-3p in hepatic tissues (Figure 1G). Here again, a difference in the age of the db/db mice (8 weeks vs. 12 weeks of age) might be the cause of these discrepancies. It is noteworthy that db/db mice become obese due to hyperphagia induced by leptin receptor deficiency and diabetes develops in a highly strain-dependent manner secondary to genetic mutations poorly relevant in the human disease [34]. The mechanisms leading to obesity and insulin resistance in db/db mice are therefore different from those involved in C57BL/6 mice fed an obesogenic diet, which, in our opinion, reflect more faithfully obesity-related metabolic disorders of the hepatic glucose and lipid metabolism in human.

The method used to decrease miR-22-3p in the mouse model could be yet another reason leading to the different outcomes on obesity observed in the db/db mouse model, as compared to our study (miR-22-3p antagomir injection in a genetically-obese model vs. genetic ablation of *MiR22* in a high-fat diet-induced obesity model). Dichotomous effects between antagomirs and genetic deletion of miRNAs in mice have been previously described to result in different outcomes in vivo. This was typically the case for studies investigating the role of the stress miRNA, miR-21-5p, in liver or in heart diseases [22,35]. Whether the methodology used to silence miR-22-3p represents a confounding factor to understand its role similarly to what has been found for miR-21-5p, remains however to be established. Although previous studies have suggested that miR-22 inhibition might represent a potential therapeutic strategy for NAFLD and insulin resistance, our study calls for caution, since outcomes of miR-22 inhibition on metabolic disorders seem to be highly dependent on the experimental model and context. Future studies assessing the outcomes of miR-22 deletion specifically in hepatocytes, in adipose tissue or in immune cells for example, should bring further important insights delineating the specific role of this complex miRNA in the different tissues contributing significantly to NAFLD.

As in the case of miR-21-5p [22,36], high levels of miR-22-3p are expressed in the liver and other tissues, but its genetic deletion in mice under normal breeding conditions leads to no significant phenotypic alterations, except the development of weak glucose intolerance. This observation was confirmed in a recent study silencing miR-22 via adenovirus construct [8]. A significant impact of *Mir22* deficiency on metabolic functions was observed only when mice were fed with an obesogenic diet. This indicates that miR-22-3p behaves as a stress miRNA, which might be expressed in tissues, but whose activity might be refrained depending on the environmental conditions. In this regard, the activity of miR-22-3p was previously described to be negatively regulated by «sponge» factors, such as long non-coding RNAs (lncRNAs) (Appendix A), which include metastasis associated lung adenocarcinoma transcript 1 (*MALAT1* [37]), myocardial infarction-associated transcript (*MIAT* [38]) and the glucosylceramidase pseudogene 1 (*GBAP1*) [39]. In addition, many other lncRNAs are predicted to bind to miR-22-3p and thus may represent key regulators of miR-22-3p activity. The existence of such regulatory mechanisms suggests that evaluating only the expression of miR-22 is not fully reliable to assess its pathophysiological role. Indeed, we cannot exclude that abnormal expression of negative regulators of miR-22-3p, (e.g., lncRNA, RNA-binding proteins) impairs its activity, but not necessarily its expression in pathological conditions. Consistent with this concept, several lncRNAs known to sponge miR-22-3p are upregulated in human/mice NAFLD models (Appendix A), indicating that even if miR-22 expression is not downregulated, its activity might be. Further studies are now required to determine not only the expression, but also the bioavailability and activity of miRNAs, e.g., miR-22 in the liver, in order to evaluate their potential as biomarkers, as well as their pathophysiological functional relevance.

The regulation of major effectors of glycolysis in hepatocytes from miR-22KO mice likely represents an important mechanism restraining diet-induced steatosis development, since glycolysis can fuel de novo lipogenesis by over-producing pyruvate and acetyl-CoA. The in vivo effect of miR-22-3p inhibition on hepatic homeostasis might not be uniquely related to these potential miR-22-3p specific targets. Indeed, 54 cellular factors involved in various metabolic processes were found upregulated in our proteomic analysis, suggesting that exacerbation of steatosis in miR-22KO mice originates from pleiotropic alterations in hepatic metabolic homeostasis. For example, increased expression of factors promoting lipid uptake in the liver of miR-22KO mice is likely also an important mechanism contributing to the abnormal accumulation of fat in the liver of these mice. This mechanism is further relevant in the human pathology, since it constitutes one of the major pathways upregulated in the liver of obese patients suffering from steatosis [40]. Thus, the effect of *MiR22* deficiency on hepatic and whole-body metabolism in fine is likely due to the deregulation of a myriad of miR-22 direct or indirect targets involved in various aspects of the metabolism, rather than one single gene candidate that would convey this phenotype. This claim is in line with the huge number of predicted and validated targets of miR-22-3p (Figure 1A). However, all potential or previously validated miR-22 targets are not all upregulated when miR-22-3p is absent. Indeed, miRNAs exert a fine-tuned repression, dependent on the cell type-specific context. The characteristic molecular traits of a given cell type can enhance/decrease the activity of miR-22-3p and thus guide its functions in a cell-specific manner [41,42]. Indeed, all miR-22-3p targets are not equally expressed in all cell types, therefore being either unavailable for repression or, on the contrary, titrating available miR-22 copies away from other targets [41,42,43]. The same context-specific functions are observed when cells undergo stress and activate stress-responses, which modulate the signaling pathways, affecting, in turn, the activity of the miRNA (molecular «sponges», difference in the activity of protein complexes conveying miRNA repression, etc.). Thus, the function of this molecule can be highly versatile [43].

Since miRNAs can regulate protein expression at the post-transcriptional level, independently of alterations of mRNA expression, we believe that proteomic analyses are more suitable than transcriptomic analyses to uncover miRNA targets. Our choice to perform proteomic analysis rather than transcriptomic was based on the fact that mRNAs of the target genes of miR-22 are not necessarily degraded, but the translation could still be blocked due to the binding of the miRNA. Ergo, transcriptomic analysis would not reflect this inhibition, whereas proteomic analysis can highlight the net result of this interaction by reflecting the actual translation of the target protein [43]. Although proteomic mass spectrometry analyses are quantitatively and qualitatively more relevant than Western blot analyses to assess changes in protein levels [44], it is however evident that sensitivity of proteomics experiment is not sufficient to detect all potential miR-22-3p targets upregulated in hepatic tissues of miR-22KO mice. As well, prediction algorithms to identify miRNA targets are sometimes of limited value, as shown by the high number of experimentally validated targets for specific miRNAs, e.g., miR-21-5p, that could not be predicted by bioinformatic tools [22,36]. We are, therefore, expecting to uncover other major metabolic drivers under the control of miR-22-3p in future studies. Of note, with our proteomic analysis, we found increased expression of glycolytic enzymes such as glucokinase and enolase 1, some of them being predicted or having been experimentally validated (e.g., enolase 1 [45]) as direct targets of miR-22-3p.

Finally, we have neglected in this study any potential function for the passenger strand miR-22-5p, which is also deleted in the miR-22KO mice. Although the canonical model of miRNA biogenesis proposes that the miRNA passenger strands are rapidly degraded, recent evidence suggests that they can also target specific mRNAs [46]. Currently, there are no pathophysiological functions described for miR-22-5p, however, increased levels of circulating miR-22-5p were detected in the serum of patients with acute phase myocardial infarction [47] or suffering from Huntington disease [3]. Interestingly, the hepatic expression of this strand was decreased in rats with NAFLD; nevertheless, the study did not present any functional data for this miRNA [48]. This question needs, however, to be addressed adequately in the future to definitively exclude any pathophysiological roles of miR-22-5p in the liver.

As mentioned above, the role of a single miRNA and the cellular outcomes associated with deregulations of its expression/activity are highly dependent on the cell type, stress conditions and the environmental context [18]. Supporting this concept, miR-22-3p inhibition in vivo in metabolically stressed hepatocytes or hepatic transformed cancer cells appears to affect differently important cellular processes such as glycolysis and mitochondrial respiration. Previous studies highlighted a strong downregulation of miR-22-3p in HCC [15,16], but most of the research efforts have focused on miR-22-3p’s oncogenic targets, which are upregulated in HCC. While the role of miR-22-3p-regulated oncogenes is undoubtedly relevant in HCC outcome, the metabolic impact of miR-22-3p alterations in HCC is not to be neglected, especially if miR-22-3p targeting is being considered as a therapeutic option.

Different therapeutic strategies are being currently developed to modulate miRNA expression/activity in specific tissues. In this regard, several classes of synthetic oligonucleotides inhibiting miRNAs (antimiRs) have been developed with different characteristics depending on their chemical structure, e.g., 2′-O-methoxyethyl-conjugated oligonucleotides, 2′-O-methyl-cholesterol-conjugated oligonucleotides (antagomiRs), or locked nucleic acids (LNA) chemically modified to increase their stability in body fluids. Important progress has also been made to optimize the conveyance of these oligonucleotides to specific organs such as the liver using, for example, various lipid conjugates and liposomal solutions [3,4]. However, in the case of miR-22-3p, which is decreased in HCC and classified so far as a tumor suppressor, therapeutic strategies should aim on the contrary, to rescue, and not to inhibit, miR-22-3p expression in hepatic cancer. Pharmacological supplementation of functional miR-22-3p-like oligonucleotides should have also a potential positive outcome even for pre-cancerous stages in NAFLD, based on our results presented in this study. Unfortunately, optimal and efficient in vivo delivery of synthetic miRNA mimicking oligonucleotides to specific organs is more challenging than in the case of antimiRs. Indeed, miRNA mimicking nucleotides are not prone to undergo chemical modifications able to increase for example their stability, since they need to remain functionally compatible with the silencing molecular machinery of cells in order to act as endogenous miRNAs. Future studies should therefore still develop and optimize functional mimicking oligonucleotides and their delivery to target organs before envisaging such therapeutic approaches.

Nevertheless, our data in hepatic cancer cell line Huh7 indicated that in transformed hepatic cells, rescuing miR-22-3p levels resulted in an increased glycolysis rate, as well as a decrease in mitochondrial respiration. This typically cancerous hallmark of reprogramming the cell metabolism to a more glycolytic phenotype could thus negatively impact the outcome of HCC patients, since this process provides metabolic adaptations suited for HCC progression (e.g., increased metabolites for lipid, amino acids, and nucleotide synthesis) [49]. Therefore, these findings should instruct future studies to be cautious regarding the development of a therapeutic approach involving miR-22-3p, as this miRNA presents a highly context-dependent versatility of its functions.

## Figures and Tables

**Figure 1 jpm-10-00170-f001:**
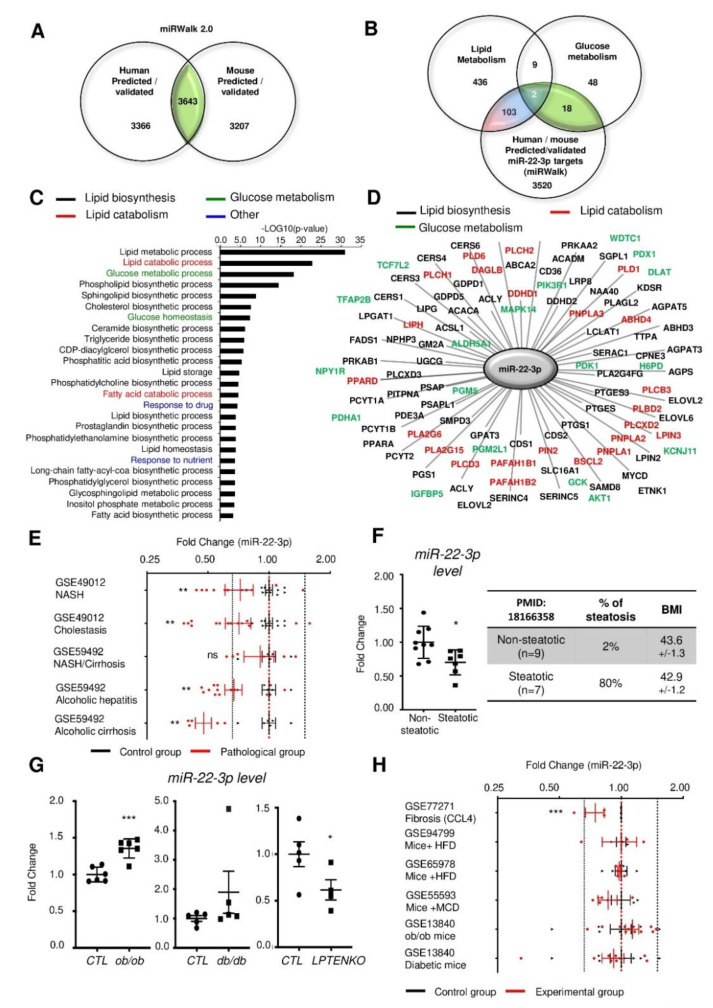
MiR-22-3p expression in the liver is not a reliable marker of non-alcoholic fatty liver disease (NAFLD) and/or non-alcoholic steatohepatitis (NASH). (**A**) Comparison of validated and predicted miR-22-3p targets in mice and humans from the miRwalk 2.0 database. (**B**) Common targets between mouse and human in Panel A are compared with a list of lipid/glucose metabolism-related genes obtained with the MetaCore^TM^ software (123 candidates). (**C**) The 123 candidates that were identified, were then subjected to a Gene Ontology analysis for biological processes. Genes enriched in lipid biosynthesis, lipid catabolism, and glucose metabolism biological processes are represented in (**D**). (**E**) Gene Expression Omnibus (GEO) datasets analyses of miR-22-3p expression in human fatty liver disease. (**F**) MiR-22-3p expression in human liver biopsies of non-steatotic (n = 9) and steatotic (n = 7) obese patients. (**G**) MiR-22-3p expression in liver tissues from 2-months old ob/ob, db/db and 4-months old LPTENKO (Liver-specific PTENKO mice) and their respective controls. (**H**) GEO dataset analyses of miR-22-3p expression in mouse fatty liver disease models (CCL4—carbon tetrachloride, HFD—high fat diet, MCD—methionine choline-deficient diet). Data represent the means ± standard error of the mean (SEM). * *p* < 0.05, ** *p* < 0.01, *** *p* < 0.001 compared with controls (Student t-test).

**Figure 2 jpm-10-00170-f002:**
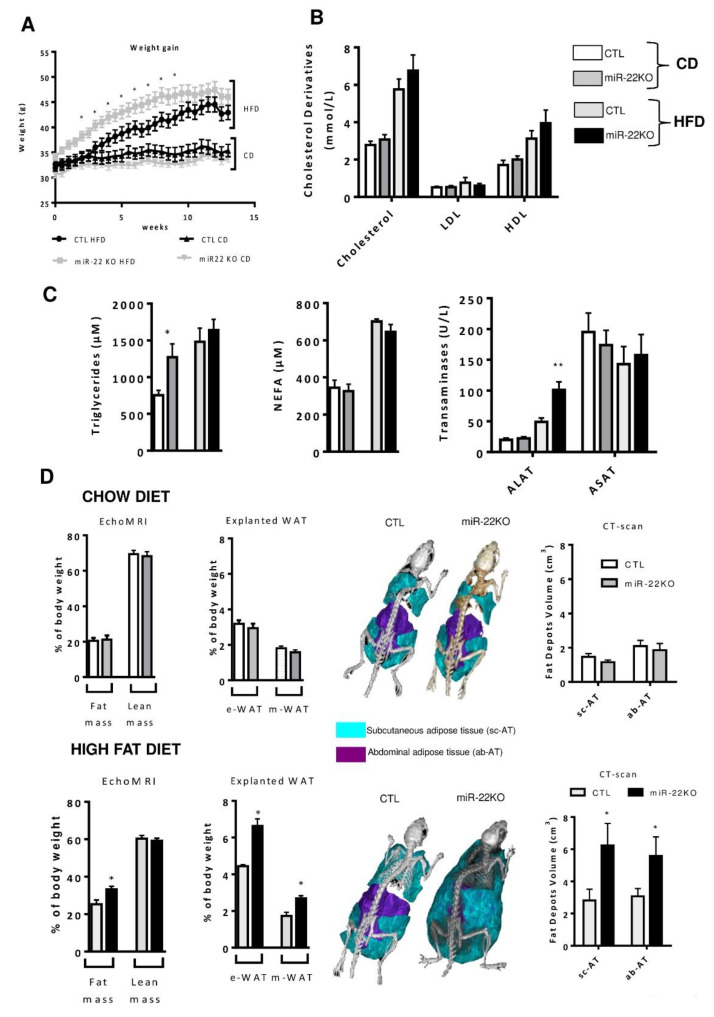
*Mir22* deficiency promotes body weight gain and adiposity in mice fed an obesogenic diet. Two-months-old control (CTL) and miR-22KO mice were fed chow (CD) or high-fat containing diet (HFD) for 12 weeks. Panel (**A**) represents weight gain of CTL and miR-22KO mice fed CD or HFD. (**B**) Plasma cholesterol, low density lipoprotein (LDL) and high density lipoprotein (HDL) levels in CTL and miR-22KO mice fed CD or HFD. (**C**) Non-esterified fatty acids (NEFA), triglycerides and plasma transaminase (ALAT and ASAT) levels in CTL and miR-22KO mice fed CD or HFD. (**D**) EchoMRI analyses of fat and lean body mass (left panels), weights of explanted epidydimal (eWAT) and mesenteric (mWAT) white adipose depots (middle panels), and three-dimensional reconstitution and volume of CT-scan analyses of fat depots (sub-cutaneous, sc-AT; abdominal, ab-AT) in CTL and miR-22KO mice fed a CD or HFD for 12 weeks. (n = 5–12 mice per group and data are represented as means ± SEM). * *p* < 0.05, ** *p* < 0.01, *** *p* < 0.001, Panels A–C were analyzed with student t-test, panel D was analyzed with ANOVA).

**Figure 3 jpm-10-00170-f003:**
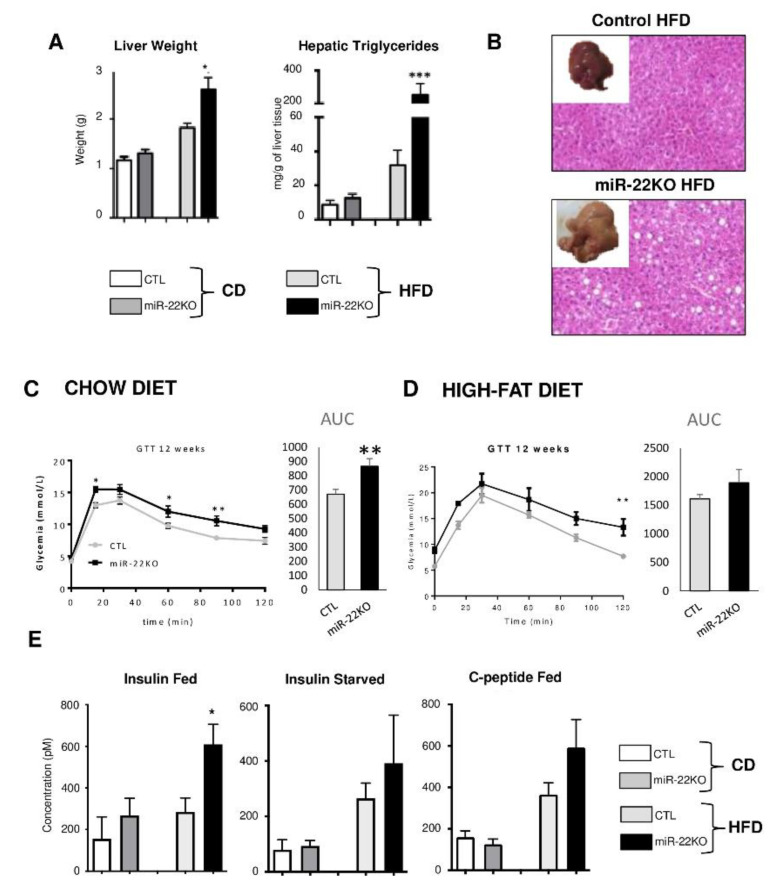
*Mir22* deficiency fosters obesity-associated hepatomegaly, hepatic steatosis and glucose intolerance. Two-months-old CTL and miR-22KO mice were fed CD and HFD for 12 weeks: (**A**) liver weights and hepatic triglycerides contents. (**B**) Representative liver anatomies and hematoxylin and eosin (H&E) staining of liver sections from CTL and miR-22KO mice fed HFD. (**C**,**D**) Glucose tolerance test (GTT) and calculated areas under the curve (AUC) in CTL and miR-22KO mice fed CD or HFD for 12 weeks (n = 5–10 per group, data are represented as means ± SEM). (**E**) Insulinemia and C-peptide plasma concentration in fed and starved mice. (n = 3–10 per group, data are means ± SEM). Statistical tests were performed: two-way ANOVA * *p* < 0.05, ** *p* < 0.01, *** *p* < 0.001.

**Figure 4 jpm-10-00170-f004:**
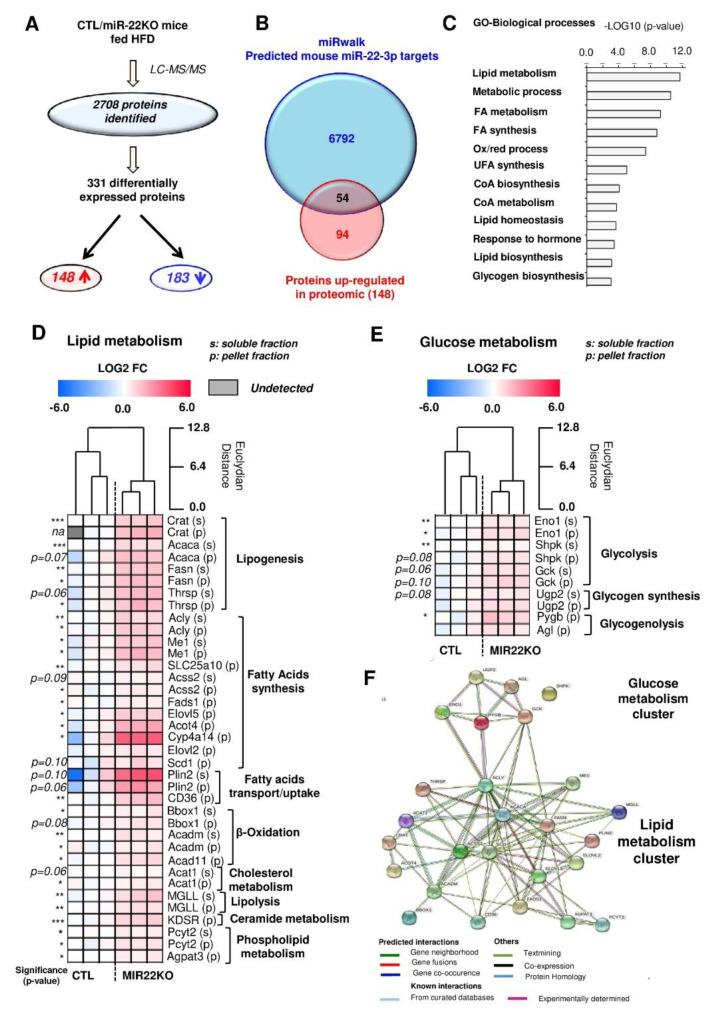
Hepatic proteomic analysis highlights important metabolic deregulations in miR-22KO mice. Two-months-old CTL and miR-22KO mice were fed high fat diet (HFD) for 12 weeks before processing explanted liver tissues for mass spectrometry analysis. (**A**) Workflow of the proteomic data analysis. (**B**) Cross-comparison of hepatic proteins up-regulated by *Mir22* deficiency with predicted mouse miR-22-3p targets according to miRwalk 2.0. (**C**) Gene-Ontology (GO) enrichment analysis for biological processes performed on the common candidates from panel B (54 candidates). 30 candidates were identified as key factors involved in lipid and glucose metabolism (based on Uniprot annotation) and potentially regulated directly by miR-22-3p. Relative protein level variations between CTL and miR-22KO mice are represented in heatmaps (**D**,**E**). (**F**) Graphic representation of the interaction of the proteins (glucose and lipid metabolism), whose mRNAs are predicted to be miR-22 targets and found upregulated in miR-22KO livers of mice challenged with 12 weeks of HFD. The representation was done via the STRING database software. (n = 5–10 per group and data are mean ± SEM, Panels D and E were analyzed via Student T test). * *p* < 0.05, ** *p* < 0.01, *** *p* < 0.001.

**Figure 5 jpm-10-00170-f005:**
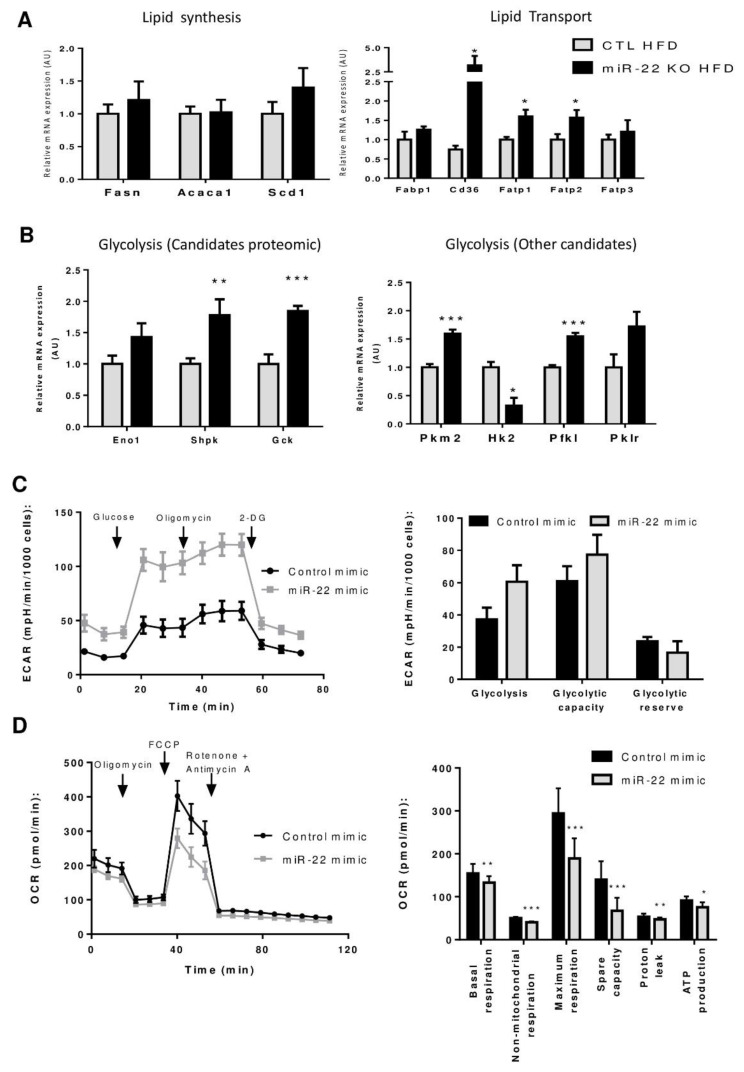
MiR-22-3p-dependent glucose/lipid metabolic activity. (**A**) RT-qPCR analyses of major effectors of lipid synthesis and transport in the livers of mice challenged 12 weeks with HFD. (**B**) RT-qPCR analyses of major effectors of glycolysis, predicted (*Eno1, Shpk,* and *Gck*) or not predicted as targets of miR-22-3p, in CTL and miR-22KO livers of mice challenged 12 weeks with HFD. For (**A**) and (**B**), data are mean ± SEM, n = 6–10 per group. * *p* < 0.05, ** *p* < 0.01, *** *p* < 0.001, compared with controls (Student t-test). (**C**) Seahorse glycolytic rate measurement (GlycoStress test) and (**D**) mitochondrial respiration rates (MitoStress test) in Huh7 human hepatic cancer cell line, transfected or not with 20 nM of miR-22-3p oligonucleotides for 48 h (n = 3). Data are represented as mean ± SEM. * *p* < 0.05, ** *p* < 0.01, *** *p* < 0.001, compared with controls (Student t-test).

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
