# Peer review of "Genetic Ablation of MiR-22 Fosters Diet-Induced Obesity and NAFLD Development"

_jpm, 2020, doi:10.3390/jpm10040170_

Round 1
Reviewer 1 Report
In this manuscript, the authors investigate the pathophysiological role of the miR-22 in the development of NAFLD and HCC. Based on the analysis of transcriptomic GEO datasets, the authors state that miR-22-3p is not a reliable marker for liver metabolic disorders. Yet, the deletion of miR-22-3p in mice induce a significant aggravation of high-fat-diet-induced, fat mass gain, glucose intolerance and liver steatosis. Moreover, hepatic proteomic analysis highlights glycolysis and lipid uptake deregulations in miR-22KO mice. Finally, the authors observed that miR-22 expression in hepatic cancer cells such triggers a classical Warburg-like metabolic switch. Overall, the manuscript is well written and present an interesting insight in energy metabolism. I suggest that a few minor concerns should be address before acceptance.
Introduction
It has been recently suggest a new terminology that more accurately reflects the pathogenesis of nonalcoholic fatty liver disease (NAFLD): metabolic-dysfunction-associated fatty liver disease (MAFLD). Did you deliberately use the terminology NAFLD ?
Methods
Are the Western Blot were performed in duplicate? How did you normalized the relative abundance of a specific protein?
What was the composition of the high-fat containing diet? Is the diet recognized to induce obesity-associated steatosis.
Results
The results revealed that miR-22KO mice fed with HFD exhibit a significant aggravation of glucose tolerance. As you did not shown an alteration of insulin pathway in the eWAT, why did you not investigate skeletal muscle or, to a lesser extent, liver insulin pathway? You did explain why proteomics might more suitable to detect the change in protein expression but I don’t understand why Western Blot was only performed in eWAT.
The quality of the figure is poor but might be due a problem during the formatting of the pdf files.
Discussion
L367 In this paragraph, you discuss about the limitations of genetic models and specific deletion of miRNA. I wonder if a model of hepatic miR-22 knockout has already been generated and studied in the literature. For example, Jiang et al. demonstrates that miR-221/222 is crucial for the regulation of lipid metabolism, inflammation and fibrosis in the liver (Jiang et al. 2018 EBioMedicine).
You state at the end of the manuscript that development of therapeutic approaches should be aware of the pathologic context. Can the authors briefly provides the actual strategies to modulate miRNA expression (i.e.: LNA‐mediated suppression of miRNA…).
Author Response
REVIEWER 1
In this manuscript, the authors investigate the pathophysiological role of the miR-22 in the development of NAFLD and HCC. Based on the analysis of transcriptomic GEO datasets, the authors state that miR-22-3p is not a reliable marker for liver metabolic disorders. Yet, the deletion of miR-22-3p in mice induce a significant aggravation of high-fat-diet-induced, fat mass gain, glucose intolerance and liver steatosis. Moreover, hepatic proteomic analysis highlights glycolysis and lipid uptake deregulations in miR-22KO mice. Finally, the authors observed that miR-22 expression in hepatic cancer cells such triggers a classical Warburg-like metabolic switch. Overall, the manuscript is well written and present an interesting insight in energy metabolism . I suggest that a few minor concerns should be address before acceptance.
We would like to thank the reviewer for taking the time to review out manuscript and for providing constructive suggestions to improve it. We have addressed point by point each comment as presented below. Modifications made in the manuscript are marked in italic.
Introduction
It has been recently suggested a new terminology that more accurately reflects the pathogenesis of nonalcoholic fatty liver disease (NAFLD): metabolic-dysfunction-associated fatty liver disease (MAFLD). Did you deliberately use the terminology NAFLD ?
We were indeed aware of the new terminology proposed by some authors and pointed by the Reviewer. We, however, have chosen to use the term NAFLD as in the hepatology field this acronym is widely used for this particular pathology, while MAFLD is still not very familiar to the hepatology community. As the major scientific/medical societies related to liver diseases, i.e. European Association for the Study of the Liver (EASL) and American Association for the Study of Liver Diseases (AASLD), still haven’t officially accepted MAFLD as a term to replace NAFLD, we are reluctant to use it in our manuscript. Nevertheless, this point raised by the reviewer is relevant and thus we modified our introduction to mention this new terminology, which will likely be adopted in the future, as followed: L63-L66 – “Obesity and the metabolic syndrome are associated with a spectrum of hepatic pathologies classically defined as Non-Alcoholic Fatty Liver Disease (NAFLD) or, as suggested more recently as Metabolic Dysfunction-Associated Fatty Liver Disease (MAFLD). These hepatic metabolic disorders start with an excessive accumulation of lipids in the hepatocytes (steatosis)….”
Methods
Are the Western Blot performed in duplicate? How did you normalized the relative abundance of a specific protein?
We apologize for not having been clear regarding this important information in our manuscript. The western blot in supplementary figure 8 was performed with samples coming from 5 individual control mice and 6 individual miR22KO mice (each line represents one single mouse) and quantifications are related to the representative WB provided in Fig S8. For info, these WB were performed in two different cohorts of mice with similar results, but data from only one cohort are presented here). Insulin stimulation in Fig S9, were done in a cohort of 6 control and 6 miR22KO mice fed the obesogenic diet and treated or not with insulin 40 minutes before sacrifice. Each line of the WB in Fig. S9 thus represent explanted eWAT from a single mouse.
For figure S8, the protein contents of CD36, ACC, FASN and SCD1 were reported to the β-actin content in each mouse, and the mean ratio (mean +/-SEM) was calculated for the group. For supplementary figure S9, the content of phospho-IR was reported to the total IR content and the phosphoThr-AKT and phosphoSer-AKT were reported to the total AKT content in each mouse, and subsequently the mean for the group was calculated (mean +/-SEM). The following experimental details were added to clarify these points in the supplementary figure legends:
S8 – “Representative Western blots (n=5 for control mice and n=for miR22KO mice) and quantifications (n=5-6 mice per group, data are means ± SEM) … The protein contents of CD36, ACC, FASN and SCD1 were reported to the β-actin content in each mouse….”
S9 – “…(n=2-3 different mice for control groups treated or not with insulin, and n=3 different mice for miR-22KOmice treated or not with insulin, data are means±SEM). The content of phospho-IR was reported to the total IR content and the phosphoThr-AKT and phosphoSer-AKT were reported to the total AKT content in each mouse….”
What was the composition of the high-fat containing diet? Is the diet recognized to induce obesity-associated steatosis.
We apologize for not having clearly indicated where to find the references and information for the diets used for animal experiments in the main text of this manuscript. The provider and the references for the standard and the high-fat-containing diet were actually stated in the supplementary table 3. The high-fat diet was purchased from ssniff Spezialdiäten GmbH (Soest, Germany). The reference for the diet is E15741-34, EF D12492 (II) mod. (60 kJ% fat, 21 kJ % carbohydrates and 19 kJ% protein). The precise molecular content can be found at: https://www.bio-services.nl/cms/files/Groep2_loboratoriumvoeders/10_catalogue_EF_new_Experimentele_voeders.pdf .
We have previously used the same diet to investigate in vivo in mice the role of miR-21 in obesity-associated hepatic steatosis (Calo et al., Gut, 2016, https://pubmed.ncbi.nlm.nih.gov/27222533/ ), and other research groups also reported a diet-induced obesity with this same feeding regime (e.g. PMID: 26077714, PMID: 31776435).
This info has now been clarified in the Table 3 of supplementary materials and Methods and references to the Table S3 and our previous publication have been added to the revised text of our manuscript “…(HFD, 12 weeks, see Table S3) [26].”
Results
The results revealed that miR-22KO mice fed with HFD exhibit a significant aggravation of glucose tolerance. As you did not shown an alteration of insulin pathway in the eWAT, why did you not investigate skeletal muscle or, to a lesser extent, liver insulin pathway? You did explain why proteomics might more suitable to detect the change in protein expression but I don’t understand why Western Blot was only performed in eWAT.
Since miR22KO mice display phenotypic alterations mostly in lipogenic tissues (white adipose tissue and liver) we focused our investigations on these two tissues. The increased pic of plasma glucose at 15 min in the GTT (Fig.3C, GTT in mice under chow diet) with then a similar rate of glucose disappearance further suggested that miR-22KO mice have mostly an alteration of hepatic glucose output and not of muscle glucose uptake. We have now incorporated this comment in our data description of these GTT experiment in the result section line 290-293 as followed:
"The increased pic of plasma glucose at 15 min in miR-22KO mice under chow diet (Fig. 3C), with then a similar rate, as compared to CTL mice, of glucose disappearance further suggests that miR-22-deficiency leads to alterations in hepatic glucose output”.
Based on this interpretation of our data, we therefore assessed also insulin signaling in the liver (but not in muscles). Similar than in the adipose tissue, we actually did not observe any significant differences in the initial steps of hepatic insulin signaling suggesting that defects likely occur downstream of Akt or in insulin-independent processes. We did not add these data in the manuscript because we encountered a problem with the hepatic response to insulin in one control mouse for an unknown reason (see figure below - only 2 samples for WT mice treated with insulin). Since it was not ethically justified to redo an obesogenic diet in an additional group of mice to further confirm this negative data, we did not add them as supplementary results together with the adipose tissue. We however provide these data for the reviewer only here below.
As we previously reported several times, WB analyses are useful to rapidly and easily assess the expression (or activity) of few particular and well characterized enzymes having major roles in a given process. Based on the observed phenotype of the adipose tissue (increased adiposity), we could indeed expect alterations of key lipogenic enzymes predicted to be targeted by miR-22, which was indeed the case as demonstrated by WB in the supplementary figure S8. However, when investigating the potential role of miR-22 in the liver (which was the major goal of this study), we decided to deepen and broaden our analyses of the impact of miR-22 deficiency in the liver metabolism. We therefore performed a high-throughput and unbiased approach, i.e. analyses of changes in the proteome by mass spectrometry, to identified major changes in the proteome and potential miR-22 targets. Based on expert opinion in the field, proteomic analyses are more reliable in terms of identification and quantification of proteins when these latter are detectable, as compared to Western blot analyses, which are highly dependent of the antibody specificity, sample processing, etc. Therefore, further analyses by Western blot approaches in the liver to identify/confirm potential miR-22 target affected in hepatic tissues were not required. Of note also, our proteomic analyses confirmed in a more reliable and quantitative manner in the liver, changes that we observed also in the adipose tissue (e.g. Fasn, Acaca and Scd1, see Fig S8 and Fig 4D). Based on the already extensive data obtained with the liver in this study, we judge unnecessary to perform also expensive and complex proteomic analyses in the adipose tissue.
We hope that our experimental strategy regarding these two key points raised by the reviewer , is now more clear and that the reviewer will agree with the approaches that we have undertaken.
The quality of the figure is poor but might be due a problem during the formatting of the pdf files.
We thank the reviewer to point out this issue. While the PDF files of the figures submitted separately were of good quality, the same figures inserted in the text had lower resolution. We have now replaced all images with new ones with a 300 dpi resolution.
Discussion
L467 In this paragraph, you discuss about the limitations of genetic models and specific deletion of miRNA. I wonder if a model of hepatic miR-22 knockout has already been generated and studied in the literature. For example, Jiang et al. demonstrates that miR-221/222 is crucial for the regulation of lipid metabolism, inflammation and fibrosis in the liver (Jiang et al. 2018 EBioMedicine).
This is a very important point that the reviewer is raising. To our best knowledge, there is no hepatocyte-specific knock-out model of miR-22, which has been generated and described in the literature. In the reference that the reviewer cited, hepatic deletion of miR-221/222 was performed in mice, and miR-221/222 was then re-expressed in these mice through adenoviruses transduction to support a role for these microRNAs in the liver. We believe that the most pertinent in vivo approaches to understand the role of miRNAs should privilege strategies based on genetic deletions of miRNA of interest. Knockout-based strategies have indeed the advantage to prevent potential off-target and non-physiological effects of overexpressed miRNAs (see Gjorgjieva et al Gut, 2019), which cannot undergo a relevant pathophysiological control of their expression/activity. In this regard, we agree with the reviewer that future studies examining tissues specific deletions of miR-22 in the liver, the adipose tissue or even immune cells should provide key insights about how miR-22 orchestrate metabolism in these different but tightly connected tissues. The importance of future studies assessing the outcomes of tissue-specific miR-22 knockout mice in NAFLD has now been outlined in the discussion as followed: Lines 477-480 - Future studies assessing the outcomes of miR-22 deletion specifically in hepatocytes, in adipose tissue or in immune cells for example, should bring further important insights delineating the specific role of this complex microRNA in the different tissues contributing significantly to NAFLD.Potential issues with overexpression of miRNAs through expression of genetic constructs have been added in the discussion.
You state at the end of the manuscript that development of therapeutic approaches should be aware of the pathologic context. Can the authors briefly provide the actual strategies to modulate miRNA expression (i.e.: LNA‐mediated suppression of miRNA…).
We thank the reviewer for this comment and agree that it is worth to briefly discuss this topic in the discussion. We thus modified substantially the last paragraph of the discussion as shown below:
Different therapeutic strategies are being currently developed to modulate miRNA expression/activity in specific tissues. In this regard, several classes of synthetic oligonucleotides inhibiting miRNAs (antimiRs) have been developped with different characteristics depending on their chemical structure, e.g, 2′-O-methoxyethyl-conjugated oligonucleotides, 2′-O-methyl-cholesterol-conjugated oligonucleotides (antagomiRs)), or locked nucleic acid (LNA) chemically modified to increase their stability in body fluids. Important progresses have also been made to optimize the conveyance of these oligonucleotides to specific organs such as the liver using for example various lipid conjugates and liposomal solutions [3,4]. However, in the case of miR-22-3p, which is decreased in HCC and classified so far as a tumor suppressor, therapeutic strategies should aim on the contrary to rescue, and not to inhibit, miR-22-3p expression in hepatic cancer. Pharmacological supplementation of functional miR-22-3p-like oligonucleotides should have also a potential positive outcome even for pre-cancerous stages in NAFLD, based on our results presented in this study. Unfortunately, optimal and efficient in vivo delivery of synthetic miRNA mimicking oligonucleotides to specific organs is more challenging than in the case of antimiRs. Indeed, miRNA mimicking nucleotides are not prone to undergo chemical modifications able to increase for example their stability, since they need to remain functionnally compatible with the silencing molecular machinery of cells in order to act as endogenous miRNAs. Futur studies should therefore still develop and optimize functional mimicking oligonucleotides and their delivery to target organs before envisaging such therapeutic approach.

Reviewer 2 Report
In this manuscript Gjorgjieva et al. presented novel insights on the miR-22-3p role in obesity and NAFLD.
They used miR-22-3p-KO mice, as well as transformed hepatic cells as models. MiR-22 deletion was found to promote hepatic glycolysis and lipid uptake, whereas miR-22-3p expression in hepatic cancer cells surprisingly increased glycolytic rate.
The manuscript is overall clear and well-written, it requires minor English language editing and spell check. I’d particularly like to emphasize the Discussion in which the authors very nicely tried to give possible explanations for discrepancies in the results shown.
Some minor points:
- Genes names should be written in italics;
- Every time new gene or protein is mentioned, it should be defined (see lines 97, 284, 290, 296, 299, 412);
- “In vivo” should not be in italics, according to the journal’s rules;
- Lines 168-172 - S3 figure shows H&E staining only, there is no miR-22-3p staining, so these conclusions are based on what?
- Line 219 - In the Figure S7 there are also fuel oxidation and water intake that are not mentioned in the text at all;
- Lines 234 – 5 – ALAT and ASAT values were measured in plasma or serum, it is not clear?
- Line 247 – Figure S7 is wrongly cited here;
- Lines 335 – 9 - the results of the Seahorse analysis described do not really match to what is shown in the Figure 5D; it would seem that the only significant is the maximal respiration rate, upon FCCP addition.
- Figure S10 – better quality images should be shown;
- Figure S13B – number of both predicted and validated lncRNAs can not be higher than the number of validated ones.
Author Response
REVIEWER 2
In this manuscript Gjorgjieva et al. presented novel insights on the miR-22-3p role in obesity and NAFLD.
They used miR-22-3p-KO mice, as well as transformed hepatic cells as models. MiR-22 deletion was found to promote hepatic glycolysis and lipid uptake, whereas miR-22-3p expression in hepatic cancer cells surprisingly increased glycolytic rate.
The manuscript is overall clear and well-written, it requires minor English language editing and spell check. I’d particularly like to emphasize the Discussion in which the authors very nicely tried to give possible explanations for discrepancies in the results shown.
We would like to thank the reviewer for taking the time to review out manuscript and for providing us with kind words and constructive suggestions to improve it. We have addressed point by point each comment as presented below. Modifications made in the manuscript are marked in italic.
Some minor points:
- Genes names should be written in italics;
As recommended by the reviewer, we went through the manuscript and we corrected all inappropriate nomenclature for the gene names.
- Every time new gene or protein is mentioned, it should be defined (see lines 97, 284, 290, 296, 299, 412);
We have now defined all the names of these genes / proteins in the manuscript.
- “In vivo” should not be in italics, according to the journal’s rules;
As suggested, we have removed the italic form for in vivo in the text.
- Lines 168-172 - S3 figure shows H&E staining only, there is no miR-22-3p staining, so these conclusions are based on what?
We are sorry for not being clear enough in these figures (Fig 1G and S3). Indeed, the hepatic expression of miR-22-3p measured by RT-qPCR in ob/ob, db/db and LPTENKO mice is represented in Fig.1G. The H&E staining in supplementary figure S3 corresponds to the histology of the livers of these 3 groups of mice from Fig.1G. We added this supplementary figure 3 for the reader to confirm the presence of steatosis in these 3 mouse models.
In the figure legend for supplementary figure 3 we now have added the following indication: The expression of miR-22-3p in the livers of these mice is presented in figure 1G.
In the main text (in the appropriate results section) we have clarified: (see histological characterization of ob/ob, db/db and LPTENKO mice in Fig. S3).
- Line 219 - In the Figure S7 there are also fuel oxidation and water intake that are not mentioned in the text at all;
We apologize for not having commented these data in the Results section. We have now corrected our description in the results section as suggested and added water intake and fuel oxidation, as indicated below:
“Finally, fatness of miR-22KO mice under HFD was not linked to a significant decrease in energy expenditure, as indicated by indirect calorimetry evaluating the respiratory exchange rate, food consumption, water consumption, fuel oxidation and locomotor activity (Figure S7).”
- Lines 234 – 5 – ALAT and ASAT values were measured in plasma or serum, it is not clear?
ALAT and ASAT were measured in plasma. We corrected our figure legend.
- Line 247 – Figure S7 is wrongly cited here;
We apologize for this mistake. Indeed, we meant to cite supplementary figure S10 and not S7.
The line now states: “However, when mice were fed 12 weeks with a HFD, miR-22KO mice developed strong hepatomegaly and extended hepatic steatosis, whereas CTL mice developed only a modest increase in liver weight and steatosis under these conditions (Figure 3A-B and S10).”
- Lines 335 – 9 - the results of the Seahorse analysis described do not really match to what is shown in the Figure 5D; it would seem that the only significant is the maximal respiration rate, upon FCCP addition.
Indeed, the reviewer is right, as we did not reach significance with the GlycoStress analysis when calculating overall glycolysis and glycolytic capacity. We thus changed the sentence describing these data in the result section of figure 5:
“Intriguingly, both glycolysis rate and capacity in cells expressing miR-22 mimics tend to increase (Figure 5C) with no changes in the mRNA expression of the glycolytic genes Eno1 and Shpk (Figure S12C), in contrast to what we observed in non-tumoural but steatotic livers of miR-22KO mice fed HFD (Figure 5B).”
- Figure S10 – better quality images should be shown;
We have added the same images with an increased quality in the figure S10.
- Figure S13B – number of both predicted and validated lncRNAs can not be higher than the number of validated ones.
We apologize for not being explicit enough in the description of supplementary figure 13B. We have thus redrawn the figure as shown below and we added an explanation to the figure legend S13B as followed:
Supplementary figure S13. Validated lncRNA targeting miR-22-3p (miRwalk/Literature)
(A) A list of validated long-non-coding RNA (lncRNA) targeting miR-22-3p was made by using miRWalk 2.0 database and a literature screening. lncRNA predicted by miRWalk 2.0 are indicated with a *. The corresponding PMIDs and models for each lncRNA are also indicated. (B) Comparison of Predicted and validated lncRNA binding miR-22-3p, based on miRWalk 2.0 database and literature. The total number of validated lncRNAs that interact with miR-22-3p is 12 (blue circle and S13A). Among all experimentally validated lncRNAs interacting with miR-22-3p, only nine of them (intersection of blue and green circle) were predicted by currently publicly available bioinformatic tools. (C) Expression of potential lncRNA involved in miR-22-3p regulation in various human/mouse models of NAFLD or HCC. Corresponding PMIDs for each lncRNA are indicated.
